# Further Delineation of Phenotype and Genotype of Primary Microcephaly Syndrome with Cortical Malformations Associated with Mutations in the *WDR62* Gene

**DOI:** 10.3390/genes12040594

**Published:** 2021-04-19

**Authors:** Ryszard Slezak, Robert Smigiel, Ewa Obersztyn, Agnieszka Pollak, Mateusz Dawidziuk, Wojciech Wiszniewski, Monika Bekiesinska-Figatowska, Malgorzata Rydzanicz, Rafal Ploski, Pawel Gawlinski

**Affiliations:** 1Department of Genetics, Wroclaw Medical University, 50-368 Wroclaw, Poland; 2Department of Pediatrics and Rare Disorders, Wroclaw Medical University, 50-368 Wroclaw, Poland; 3Department of Medical Genetics, Institute of Mother and Child, 01-211 Warsaw, Poland; ewa.obersztyn@imid.med.pl (E.O.); mateusz.dawidziuk@imid.med.pl (M.D.); wiszniew@ohsu.edu (W.W.); 4Department of Medical Genetics, Medical University of Warsaw, 50-368 Warsaw, Poland; Poli25@wp.pl (A.P.); rploski@wp.pl (R.P.); 5Department of Molecular and Medical Genetics, Oregon Health and Science University, Portland, OR 97239-3098, USA; 6Department of Diagnostic Imaging, Institute of Mother and Child, 01-211 Warsaw, Poland; monika.bekiesinska@imid.med.pl

**Keywords:** *WDR62* gene, intellectual disability, microcephaly, MCPH2

## Abstract

Type 2 congenital microcephaly (MCPH2) is a brain development disorder characterized by primary microcephaly with or without brain malformations. MCPH2 is caused by mutations in the *WDR62* gene. We present three new patients with MCPH2 and compound heterozygous mutations in the *WDR62* gene. In all the cases, the parents were healthy and unrelated. All children were clinically diagnosed with congenital microcephaly and retardation of motor and speech development. Sequencing results in the presented patients revealed five new variants in the *WDR62* gene (c.4273C>T, c.1711_1712insTA, c.1777_1778delGA, c.1642+2T>G, c.194T>A) and one previously described in the German population (c.2864_2867delACAG). In two of the presented cases, variants in the *SMAD4*, *DKC1*, and *ATRX* genes were also found with unknown effects on the course of the disease. Moreover, in the article we collected and compared the most common clinical symptoms, dysmorphic features, and changes in radiographic examinations of the brain observed in 120 patients with recessive primary microcephaly type 2 caused by mutations in the *WDR62* gene.

## 1. Introduction

Microcephaly Primary Hereditary (MCPH) is a group of brain development disorders, defined as a reduction in the volume of the brain and a secondary reduction in head circumference, usually without changes in brain architecture. The size of the brain is determined by the number of its neurons, and the process of their formation is related to the proper function of the centrosome and especially in mitotic spindle formation, and neuronal migration. A mitotic spindle formation disturbance does not change the functions of neurons. Until now, many genes involved in these processes have been described, and several types of MCPH have been distinguished [1]. The basic clinical picture includes microcephaly found immediately after birth, at least 2 standard deviations below normal age, relatively normal early motor milestones, intellectual disability of variable severity, speech development delay, motor deficit, and inconstant epilepsy. All the described cases were inherited autosomal recessively [2].

Type 2 congenital microcephaly (MCPH2; OMIM 604317) is caused by mutations in the *WDR62* gene (WD repeat domain 62) located at locus 19q13.12-q13.2. The gene is 50,230 bps in size and consists of 32 exons and has 9 splice isoforms. It has numerous repeats in wd40 and consists of 10 functional domains. The gene encodes the WDR62 protein of 1523 amino acids. It is a phosphoprotein associated with mitotic spindle poles during prophase to metaphase. This protein is involved in the pathway of the c-Jun N-terminal kinase. Its expression is seen in neural precursor cells and the postmitotic neurons of the developing brain and the ventricular and subventricular zone in the forebrain region. *WDR62* plays an important role in the proliferation and migration of neurons and the duplication of centrioles. The lack of this gene’s function leads to a significant decrease in the size of the cerebral cortex [2].

MCPH2 is the second most common type of hereditary microcephaly. So far, 120 cases of the disease have been described in the world. As the disease is inherited autosomal recessively, it has relatively most frequently been described in populations where there is a frequent occurrence of consanguineous marriages.

The article describes three new families in which novel mutations of the *WDR62* gene were found. In two of the presented cases, variants in the *SMAD4*, *DKC1*, and *ATRX* genes were also found with unknown effects on the course of the disease. Additionally, we have compared the most frequently described clinical symptoms, dysmorphic features, and changes in brain imaging observed in 120 patients described so far from among 64 families with heterozygous and homozygous mutations in the *WDR62* gene.

## 2. Materials and Clinical Results

### 2.1. Case1

A 4-month-old girl was born in the 38th week of normal pregnancy via cesarean section due to the longitudinal pelvic position. The child’s parents are young, healthy, and unrelated Caucasian people. The girl is their first child. The birth weight was 2680 g and length 52 cm. The Apgar score was 9 in the first minute. Head circumference noted at birth was 29.5 cm (<−2SD). The anterior fontanelle was small and closed quickly. Persistent foramen ovale (PFO) was found in an ECHO examination. Muscle tone was normal. An ultrasound examination of the head revealed a reduced mass of the brain. The milestones of development were achieved by the child with a considerable delay. The hearing and vision tests, as well as the metabolic tests (GC-MS, MS/MS, amino acids profile), were normal. TORCH infections were also excluded. At 4 months of age, she weighed 5.2 kg and was 59 cm long. Physical examination showed microcephaly, prominent occiput, and upslanted palpebral fissures (Figure 1a).

MRI of the head showed that the frontal and parietal lobes, as well as the temporal and occipital lobes in the right hemisphere of the brain exhibit polymicrogyria and closed schizencephaly is visible in this hemisphere. There is a significant asymmetry of the hemispheres in the brain with the right one much smaller than the left one and additionally hypoplasia of the corpus callosum, and a falx cerebri defect (Figure 1d–g). The EEG record showed low-voltage 5 Hz waves.

### 2.2. Case 2

A 20-month-old boy was born at term in the 40th week of the mother’s first pregnancy. His birth weight was 3230 g, and his Apgar score was 10. The head circumference noted at birth was 31 cm (−2 SD). During pregnancy, the mother felt fetal movements well. After birth, the newborn had no defects of the internal organs, apart from microcephaly. The anterior fontanelle was small and closed quickly. In the ultrasound examination of the head, agenesis of the corpus callosum was suspected. Hearing tests, ultrasound of the abdominal cavity, ultrasound of the heart, and the thyroid gland were normal. Ophthalmological examination showed no abnormalities. Brain MRI detected significant smoothing of the cortical gyruses in both hemispheres. In the physical examination at the age of 1 month, head circumference was 33 cm, and a slight increase in muscle tension was also shown. Physical examination showed microcephaly, sloping forehead, and upslanted palpebral fissures (Figure 1b). The hands and feet had no significant abnormalities. In the physical examination at the age of 14 months, the child sits alone but does not walk. His head circumference is 39 cm (>−2 SD).

### 2.3. Case 3

The proband is a boy of healthy and unrelated parents. In an ultrasonography examination in the 36th week of pregnancy, head growth was delayed by 4 weeks. The child was born at the 38th week with a weight of 2450 g, length 51 cm, and head circumference of 30 cm (−3.5 SD). Additional studies ruled out toxoplasmosis, cytomegaly, and zika virus infection. The baby started sitting up at 7 months of age, but later development was significantly delayed. At 8 months of age, spastic quadriparesis and ptosis of the left eyelid were diagnosed. At 2 years of age, the child does not walk independently and does not speak. MRI of the head revealed a reduction in the size of the cerebral hemispheres in relation to the size of the brainstem, cerebellum, and midbrain. There was polymicrogyria, the simplified gyral pattern of the cerebral cortex, pachygyria, and thickened cortex (Figure 1h–j). The corpus callosum and ventricular system were normal. At the age of 2, the child weighed 11.4 kg (−1.35 SD), was 86 cm tall (−1.33 SD), and had a head circumference of 41 cm (−6.54 SD). The examination revealed muscle hypotension, a narrowing of the bitemporal diameter, full cheeks, and coarse facial features (Figure 1c). At the age of 5, he still does not speak, shows signs of aggression and auto-aggression. His head circumference is 45 cm. The muscle tone is lowered, there are no signs of spasticity. The child has a severe intellectual disability. No seizures have been observed.

## 3. Methods and Genetic Results

### 3.1. Case 1

The whole-exome sequencing (WES) library was prepared using Agilent SureSelect All Exon V6 sample preparation kits and carried out on the Illumina NovaSeq 6000 sequencer, via 2 × 100 bp reads. Genomic data processing was based on an in-house developed pipeline, with reads aligned to the GRCh38 reference genome using BWA MEM. The Genome Analysis Toolkit (GATK) was used to identify SNPs and INDELs while variant calling was performed using the Haplotype-Caller. Variant annotation was carried out with the Ensembl Variant Effect Predictor (VEP). We identified two novel mutations in the *WDR62* gene (NM_001083961.1): splice-site (hg38, chr19:g.36084746-T>G, c.1642+2T>G, p.(?)) and frameshift (hg38, chr19:g.36089046_36089047delGA, c.1777_1778delGA, p.(Asp593Hisfs*9)). The test results were confirmed by Sanger sequencing. Family analysis revealed that variant c.1642+2T>G was inherited from an unaffected father and p. (Asp593Hisfs*9) from an unaffected mother. Both *WDR62* variants were not described in the HGMD and ClinVar clinical databases nor gnomAD population databases and are absent in the in-house Institute of Mother and Child database of >1000 Polish individuals examined with the use of WES. According to the ACMG classification [3] variant c.1642+2T>G was classified as pathogenic, and variant p.(Asp593Hisfs*9) as likely pathogenic. No basis for dual molecular diagnosis was found (Figure 2a).

### 3.2. Case 2

Routine chromosomal analysis showed a normal karyotype. Array comparative genomic hybridization (aCGH) showed a normal genomic copy number. Thus, whole-exome sequencing (WES) was applied for further molecular evaluation of Case 2.

WES was performed using the Human Core Exome Kit (Twist Bioscience, South San Francisco, CA, USA), according to the manufacturer’s instruction. The enriched library was paired end sequenced (2 × 100 bp) on the NovaSeq 6000 (Illumina, San Diego, CA, USA) to the mean depth of 145×. Raw data analysis and variants prioritization were performed as previously described (PMID: 32668698). Variants considered as causative were validated using DNA samples from the proband and proband’s parents by amplicon deep sequencing (ADS) performed with the Nextera XT Kit (Illumina) and paired-end sequenced (2 × 100 bp) on HiSeq 1500 (Illumina).WES analysis revealed two novel variants in *WDR62* gene (NM_001083961.1): missense ((hg38, chr19:g.036058796-T>A, c.194T>A, p.(Val65Glu)) and nonsense ((hg38, chr19:g.19:036104637-C>T, c.4273C>T, p.(Gln1425*)).

Both variants were independently confirmed in the proband by ADS. The performed family study showed that p.(Val65Glu) was inherited from proband’s father, while p.(Gln1425*) from the mother, which is consistent with in trans variants transmission in the autosomal recessive mode of inheritance (Figure 2b).

Both *WDR62* variants have 0 population frequency (according to gnomAD database v3.1 https://gnomad.broadinstitute.org/; accessed on 17 January 2021) and are absent in the in-house Medical University of Warsaw database of >3500 Polish individuals examined using WES. According to the ACMG classification [3], the p.(Val65Glu) was classified as a variant of unknown significance (VUS), while p.(Gln1425*) was classified as pathogenic. In silico pathogenicity scores used by VarSome [4] support the potential pathogenicity of both variants, including the combined annotation dependent depletion score (CADD) 33 for p.(Val65Glu) and CADD 46 for p.(Gln1425*) [5]. Moreover, variant p.(Val65Glu) is located in the same codon as the known pathogenic p.(Val65Met) mutation [6].

Besides compound heterozygote in the *WDR62* gene, a heterozygous missense variant in the *SMAD4* gene was identified (hg38, chr18:g.051078306-A>C, NM_005359.6, c.1498A>C, p.(Ile500Leu)). Family analysis revealed the absence of p.(Ile500Leu) in proband’s parents, thus the tested variant appeared as a de novo event. *SMAD4* p.(Ile500Leu) variant has 0 frequency in both the gnomAD (v3.1) database and the in-house WES database. According to the ACMG classification [3], the p.(Ile500Leu) was classified as likely pathogenic, which was further supported by VarSome pathogenicity scores [4], including CADD 21.4 (Appendix A). Interestingly, missense variants affecting isoleucine residue at codon 500, including p.(Ile500Met), p.(Ile500Thr), p.(Ile500Val), are known causative mutations for Myhre syndrome (MIM # 139210) [7]. However, at the present stage of Case 2 diagnostics, early age did not allow to determine the significance of the p.(Ile500Leu) variant and requires further observation. In our opinion we identified an incidental finding in the *SMAD4* gene.

### 3.3. Case 3

WES analysis was performed identically to Case 1. We identified two frameshift mutations in the *WDR62* gene (NM_001083961.1): hg38, chr19:g.36086755_36086756insTA, c.1711_1712insTA, p.(Asn571Ilefs*27) and hg38, chr19:g.36100872_36100875delACAG, c.2864_2867delACAG, p.(Asp955Alafs*112). The test results were confirmed by Sanger sequencing. Family analysis revealed that variant p.(Asn571Ilefs*27) was inherited from an unaffected mother while p.(Asp955Alafs*112) from unaffected father. Variant p.(Asn571Ilefs*27) was new and not described in the HGMD and ClinVar clinical databases but was present in population databases: dbSNP (rs773899700) and gnomAD (ALL:0.00042%). Variant p.(Asp955Alafs*112) was known and described in the HGMD clinical database (CD1312329) as disease-causing [8]. According to the ACMG classification [3], both variants were classified as likely pathogenic (Figure 2c).

Besides compound heterozygote in the *WDR62* gene in this patient, we found two X-linked hemizygous regulatory variants in *DKC1* (NM_001363.4): hg38, chrX:g.154762824C>G, c.-142C>G, p.(?) and *ATRX* (NM_000489.4): hg38, chrX:g.77600555T>C, c.5576A>G, p.(Asn1859Ser) genes. The *DKC1* c.-142C>G variant was described in the HGMD clinical database (CR011575) as disease-causing [9,10]. The *ATRX* p.(Asn1859Ser) variant was not described in the HGMD clinical database but was described in the ClinVar database (RCV000120175.1) with undefined pathogenicity. According to the ACMG classification [3] mutations in *DKC1* c.-142C>G and *ATRX* p.(Asn1859Ser) were classified as variants of uncertain significance (Appendix A).

## 4. Discussion

The most common genetically conditioned congenital microcephaly is associated with mutations of genes involved in the development of the central nervous system through the orientation of mitotic spindles, microtubule dynamics, DNA damage-response signaling, chromosome condensation mechanism, and transcriptional regulations that control the number of neurons generated by neural precursor cells [11]. The central nervous system is formed from a series of symmetrical and asymmetric cell divisions. Spindles parallel to the apical plane will produce flat, symmetrical (and proliferative) divisions, while vertical or oblique spindles will produce asymmetric (and differentiating) divisions [12]. Controlling the processes of differentiation and the proliferation of precursor cells is important for the final effect in the form of the final number of neurons and the size of the brain. The ratio of the number of cells that are proliferating to those that are differentiated plays a crucial role. If neurons are differentiated too early, this results in a decrease in the final number of neurons and leads to microcephaly. This process continues throughout gestation, as well as infancy and childhood to adulthood. The disruption of the spindle orientation favoring oblique differentiating divisions will promote neurogenesis at the expense of the expansion of the stem cell pool, leading to smaller brains. Most neurons are formed by the 20th week of pregnancy, and from that moment there is only an increase in volume and changes in the number of glial cells and synapses. The progression of head circumference is more and more visible in the following years of life and it results from the originally reduced number of neurons. Since the increase in brain volume is the highest in infancy (in the 1st year of life), in some cases it is possible that a significant delay in the occurrence of developmental milestones is not observed. Among the cases described so far in the literature, however, normal motor development (in the early neonatal period) was found in few cases. A severe motor development delay was observed in the majority of patients, which resulted in a lack of independent walking in a large proportion of patients. Microcephaly was found in all the described cases of the *WDR62* gene mutations. It was diagnosed at birth and at any time thereafter, but it was also visible in ultrasound performed prenatally around 32 weeks of pregnancy. The head circumferences of the newborns described in the literature ranged from 26 to 33 cm (−2 SD to −7 SD) in children born at term. In the later stages of development, microcephaly reached values up to −14 SD. The head circumference of the newborns described by us was between −2 and −6.5 SD.

Although the definition of isolated primary microcephaly assumes that no other anatomical changes were associated with this disorder apart from the reduction in the volume of the brain and the resulting reduction in head circumference, the observations of most of the cases described so far indicate the coexistence of additional disorders of the central nervous system and in most cases leading to CNS dysfunction associated clinical symptoms. The most common changes were the simplification of the cerebral gyrus pattern, thinned pachygyria, hypoplasia or aplasia of the corpus callosum, lissencephaly, schizencephaly, polymicrogyria, and cerebellar hypoplasia.

The incidence of the disease in boys and girls in all the MCPH2 cases described so far was similar (55%/45%). The age of the assessed patients ranged from 4 months to 59 years. Most of the children were born between 36 and 40 weeks of pregnancy. Body length and birth weight were normal in most cases. 

Of the 64 families, as many as 53 were consanguineous marriages. A total of 27 Pakistani families, 4 Saudi, 11 Turkish, 3 Indian, 2 Chinese, 1 Japanese, 1 Korean, 10 European (1 German, 1 Italian, 2 North European, 3 Polish, 1 Hispanic, 1 Romanian, 1 Caucasian), 1 French Canadian, 1 Mexican, 1 Arab, 1 Moroccan, 1 Tunisian family were described (Appendix A). Most of the reported cases come from countries that allow marriage between relatives. Complex heterozygotes were found only in 11 cases and their clinical picture did not differ significantly from those in which homozygotes were found [6,8,13,14,15,16,17,18,19,20,21,22,23,24,25,26,27,28,29,30,31,32,33,34,35].

The most frequently reported dysmorphic features were sloping forehead and asymmetric face, which were reported in 50 described patients. Ears were large, prominent, and low set. Full lips, especially the lower lip, were also frequently observed. The nasal bridge was prominent with a bulbous tip. Less frequently described features include upslanted fissures, hypertelorism, small chin, micro-and retrognathia, prominent occiput, impalpable fontanel, high palate, and short neck. In other parts of the body inverted nipples, broad thumb, and sandal gap were described.

Delayed psychomotor development was observed in most children, although in some the motor development was normal in the first months of life [28,34]. Hypotonia was described only in 9 out of 30 [30%] and spasticity or hyperreflexia in 16 out of 44 patients [37%]. The delay in development in some patients was so serious that some patients did not start walking or performing daily hygiene activities. Intellectual disability was observed in all the studied subjects. Most of the patients have severe, fewer of them moderate, and single cases, mild mental retardation. Speech delay was described in 66 out of 82 patients (80%). Significant aggressive behavior dominated in 10 patients. Epilepsy has been confirmed in 38 out of 103 described patients (37%).

MRI was performed in only 46 patients. Most observed features were polymicrogyria, a simplified gyral pattern of the cerebral cortex, pachygyria, and thickened cortex. Hypoplasia of the corpus callosum was described in 16 patients. More common features also included small hippocampus (7 patients) and open Sylvian fissure (7 patients). Rarely seen was the widening of the cerebral ventricles, broadening gyri, smooth brain surface, cerebellar hypoplasia, reduced volume of a hemisphere, prominent extra-axial cerebrospinal (CSF) spaces, white matter abnormalities, subcortical heterotopia, and reduced volume of a hemisphere. 

To date, 57 different mutations in the *WDR62* gene have been described (Appendix A). They were present in all exons, except exons 16, 18, 19, 24, 25, 26, and 32. Only the change c.1313G>A was present in more than two families [18,19,20,28]. Mutations c.193G>A [6,13], c.1194G>A, c.3936_3937insC, [20,33] c.3839_3855delCA, [17,33], c.3936dupC, [8,28] have been described in two families. One of the mutations we described was previously described in a patient in the German population. The remaining 50 mutations have been described only in individual cases. 

Researchers have looked into the association of a more severe course with additional environmental factors or coexistence of an additional mutation in another gene. Murdock described the coexistence of the *WDR62* mutation with the *GLI2* and *KIAA1598* gene mutations, Poulton described the *TBCD* mutation, and Nardello the *GPR56* mutation [6,24,25]. In these cases, it has been suggested that the more severe course of the disease may be associated with additional impairment of the function of genes involved in the formation of the cerebral cortex or neuronal migration. The role of the variant in the *SMAD4* gene described by us is unknown and requires further observation of the course of the disease in probands. Mutations in the *SMAD4* gene are associated with four different phenotypes, one of which is Myhre syndrome, which involves microcephaly, intellectual disability, and laryngeal/trachea stenosis, dysmorphia, and arthropathy. The coexistence of *SMAD4* and *WDR62* gene mutations has not been described so far. Apart from microcephaly, the remaining symptoms do not appear in the child we describe. However, due to age, it cannot be ruled out that it is too early for other symptoms. 

It is known that mutations in the *DKC1* and *ATRX* genes might cause microcephaly (in the case of *DKC1* seen in the Hoyeraal-Hreidarsson Syndrome variant only), so characteristic for mutations in *WDR62*, nevertheless the open question is whether what we have here is the real dual or triple molecular diagnosis. Most likely not, because in both publications cited above [9,10] mutation *DKC1* c.-142C>G is described in patients with dyskeratosis with no mentioned microcephaly. Moreover, our patient was below 3 years of age at investigation, whereas dyskeratosis in patients with *DKC1* mutations appears much later so the patient requires further observation. Mutation in gene *ATRX* p.(Asn1859Ser) has very low pathogenicity predictions (SIFT: Tolerated; MutationTaster: polymorphism; PolyPhen-2: benign, CADD:13,52.) and its role in the pathogenesis of microcephaly in our patients is unknown.

## 5. Conclusions

The paper presents the first description of three Polish families in which children were diagnosed with autosomal recessive microcephaly caused by mutations in the *WDR62* gene and the analysis of clinical and radiological symptoms in the cases of MCPH2 described earlier in the world. The children described by us did not differ significantly in their clinical course. We also did not observe significant differences depending on additional mutations in other genes. No differences were depending on the type of mutation (missense, splicing, deletion, and nonsense). In all patients with mutations in the *WDR62* gene, microcephaly was observed in the neonatal period, gradually worsening in the childhood period. The second element common to all the described patients was mild to severe intellectual disability. Often, there is also a delay in speech development. Dysmorphic features most often appearing in the course of the disease are not characteristic, but sloping forehead, large ears, and full lips were the most common. Also, the MRI images of the head were different, but in most cases the features of polymicrogyria, pachygyria, and hypoplasia of the corpus callosum were predominant. Due to the atypical clinical picture, the diagnosis of primary microcephaly associated with mutations in the *WDR62* gene is possible only with the use of whole-exome sequencing or a panel of selected genes.

## Figures and Tables

**Figure 1 genes-12-00594-f001:**
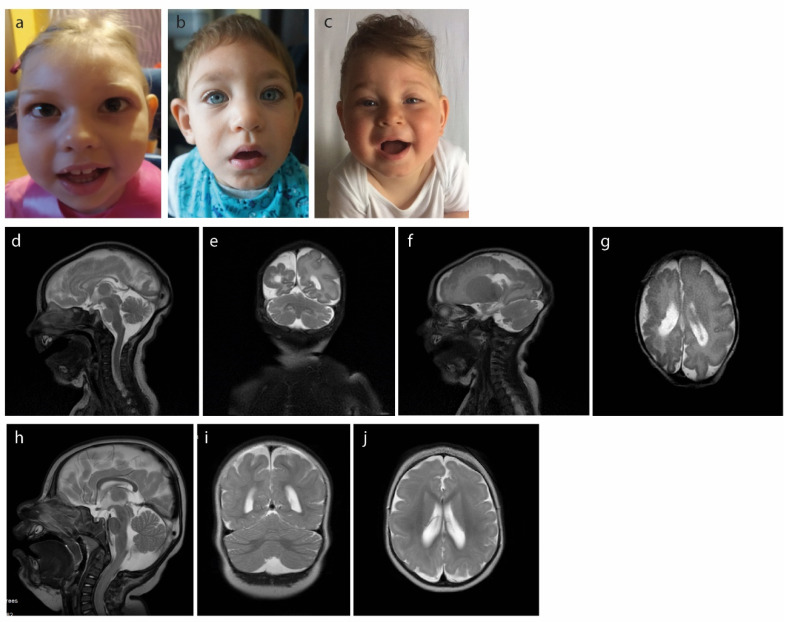
Dysmorphic facial features of individuals with *WDR62* mutations: case 1, three year old girl (**a**), case 2, two year old boy (**b**) and case 3, one year and seven days old boy (**c**). MRI analysis of case 1 with microcephaly when she was 19days old revealed a disproportionately small brain compared to the cerebellum (**d**,**e**) and disproportion of the cerebral hemispheres: R << L (**e**,**g**). Fully formed, thin corpus callosum (**d**). Polymicrogyria and closed-lip schizencephaly in the right cerebral hemisphere and simplified gyral pattern in the left one (**f**,**g**). Case 3, a 7-month-old boy with microcephaly—revealed disproportionately small brain compared to the cerebellum (**h**,**i**), dysgenesis of the corpus callosum (**h**) and simplified gyral pattern (**i**,**j**). Reduced white matter volume with posterior horn−dominant enlargement of the lateral ventricles (**j**).

**Figure 2 genes-12-00594-f002:**
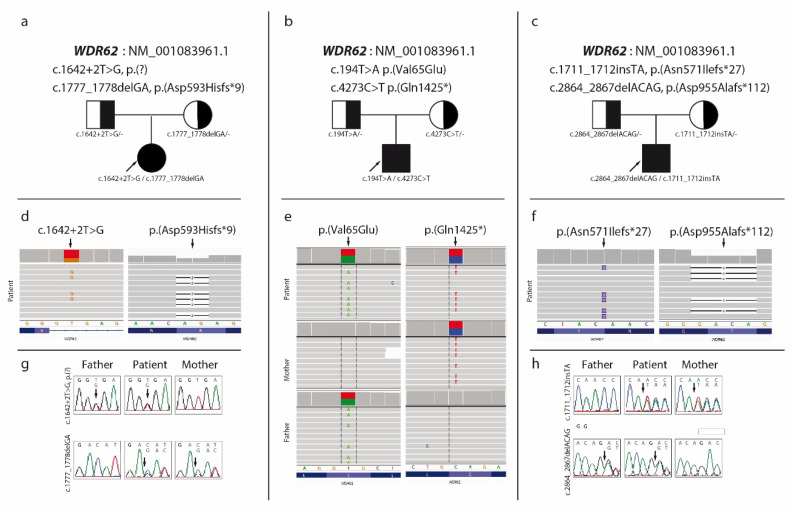
Pedigree (**a**–**c**), the Integrative Genomics Viewer (IGV) from Whole Exome Sequencing (**d**,**f**), IGV from Amplicon Deep Sequencing (**e**) and Sanger sequencing plots (**g**,**h**) showing segregation of the variants in *WDR62* in case 1 (**a**,**d**,**g**), case 2 (**b**,**e**) and case 3 (**c**,**f**,**h**). The proband is indicated with an arrow.

## Data Availability

Data is contained within the article or Appendix A.

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
