# Peer review of "Further Delineation of Phenotype and Genotype of Primary Microcephaly Syndrome with Cortical Malformations Associated with Mutations in the WDR62 Gene"

_genes, 2021, doi:10.3390/genes12040594_

Round 1

Reviewer 1 Report

Brief summary:

This manuscript describes the identification of novel WDR62 variants in three microcephaly patients and their corresponding clinical features. They largely ascribe the patient phenotypes to the mutations in WDR62 but acknowledge that mutations in three other genes may contribute to the disease phenotype in two of the patients. Finally, patient phenotypes are discussed in the greater context of all MCPH2 phenotypes.

Comments to authors:

-a lot of awkward/unclear phrasing throughout the manuscript

-protein vs. gene nomenclature rules (lines 16, 38, 51)

-there are more than 4 splice isoforms (line 37)

-are there two separate in-house Polish databases being referred to (different numbers of individuals indicated in Case 1 and 2 sections)?

-case 2 has an extra 0 at the beginning of the genomic position

-insufficient figures (brain scans? molecular/Sanger confirmation? pedigree?)

-include which patient each image corresponds to in Figure 1 and the ages at which the photos were taken

-discussion lines 249-258 are a duplicate of results lines 186-195

-citations should be included for lines 259-268

-should specify more clearly in lines 259-268 that they are summarizing WDR62 mutations

-discussion doesn’t seem very coherent/linear

-materials and methods/results sections are combined under the header of materials and methods? It would help if these sections were divided.

-incorporating the types of variants that are found in WDR62-associated microcephaly would be a beneficial addition to the introduction/discussion (since missense, splicing, and nonsense mutations were found in the study and LOF mutations fits MCPH2)

-a brief summary on how the three cases compare should be included (and then compare in the greater context of MCPH2 phenotypes)

-check annotations for variants (ie. for patient 1, should be splice-site and frameshift variant, for patient 2, should be two frameshift mutations)

-was there any evidence thus far for a composite/blended phenotype for Patient 2 who also has a SMAD4 variant?

-the DKC1 variant (CR011575) may have been reclassified; check

-be more descriptive for lines 247-248 (ie. which symptoms are not present and which symptoms are too early to observe?)

-perhaps include a table to summarize the summary of frequency of phenotypes in the discussion

Author Response

Dear Reviewer

Thank you for your very thorough analysis of the article and all valuable comments, we tried to relate to each of them with due diligence. We hope that the responses will meet with your acceptation.

The file with answers for your comments are attached .

Reviewer 2 Report

  1. The paper would be significantly improved and more impactful if the appropriate references were included in the introduction, and attributed to the specific findings in the discussion - including the description of the previous cases that have been reported.  
  2. Informal use of language, such as "tummy" should be corrected.
  3. It would be most informative for the reader to include figures of the MRIs of the 3 cases presented.  
  4. A significant finding in this paper are that these cases were not born to parents who are consanguineous, but appear to be autosomal recessive.  In addition, the child with multiple variants in additional genes is of interest to the community, and follow-up and continued study of these children will be important to understand the impact of the other variants.

Author Response

(The authors gave the same response as above.)

Round 2

Reviewer 1 Report

-the discussion reads much more clearly. Perhaps consider merging lines 276-291 into one paragraph and rewording slightly for clarity.

-Case descriptions should be in results section whereas descriptions of methods (ie. overview of clinical examination/DNA extraction and genotyping) should be in the Materials and Methods section (eg. lines 115-120, 136-143). Reorganization of these two sections is therefore recommended.

-check annotations for variants (ie. for patient 1, should be splice-site and frameshift variant, for patient 2, should be two frameshift mutations); this comment has not been updated in the text

-unclear what is meant by primary diagnosed (line 15)

-should be 29.5 cm or -6.5 SD or 13.52 (period instead of comma for measurements; ie. lines 66, 226, 203)

-indicate corresponding panels in the text for figures (ie. line 78, 106, 131, 149, 183)

-incomplete sentences (line 215, 275)

Author Response

Dear Reviewer

Thank you for your next very thorough analysis of the article and all valuable comments.

Reviewer’s comments:

-the discussion reads much more clearly. Perhaps consider merging lines 276-291 into one paragraph and rewording slightly for clarity.

Answer: The mentioned phrases were changed. The text was corrected.

-Case descriptions should be in results section whereas descriptions of methods (ie. overview of clinical examination/DNA extraction and genotyping) should be in the Materials and Methods section (eg. lines 115-120, 136-143). Reorganization of these two sections is therefore recommended.

Answer: The mentioned sections were changed.

-check annotations for variants (ie. for patient 1, should be splice-site and frameshift variant, for patient 2, should be two frameshift mutations); this comment has not been updated in the text

Answer: The mentioned phrases were changed.

-unclear what is meant by primary diagnosed (line 15)

Answer: The mentioned phrase was changed.

-should be 29.5 cm or -6.5 SD or 13.52 (period instead of comma for measurements; ie. lines 66, 226, 203)

Answer: The mentioned phrases were changed.

-indicate corresponding panels in the text for figures (ie. line 78, 106, 131, 149, 183)

Answer: The mentioned phrases were changed.

-incomplete sentences (line 215, 275)

Answer: We added the dots in the end sentences.